# Bone Mineral Density in Premenopausal Women Is Associated with the Dietary Intake of α-Tocopherol: A Cross-Sectional Study

**DOI:** 10.3390/nu11102474

**Published:** 2019-10-15

**Authors:** Tamami Odai, Masakazu Terauchi, Asuka Hirose, Kiyoko Kato, Naoyuki Miyasaka

**Affiliations:** 1Department of Obstetrics and Gynecology, Tokyo Medical and Dental University, Yushima 1-5-45, Bunkyo, Tokyo 113-8510, Japan; odycrm@tmd.ac.jp (T.O.); a-kacrm@tmd.ac.jp (A.H.); n.miyasaka.gyne@tmd.ac.jp (N.M.); 2Department of Women’s Health, Tokyo Medical and Dental University, Yushima 1-5-45, Bunkyo, Tokyo 113-8510, Japan; kiyo.crm@tmd.ac.jp

**Keywords:** osteoporosis, bone mineral density, α-tocopherol, oxidative stress, antioxidants

## Abstract

This study aimed to investigate the relationship between the consumption of various nutrients and bone mineral density (BMD) in middle-aged women. This cross-sectional survey was conducted based on the clinical records of 157 women aged 38–76. Their lumbar spine BMD was measured with dual-energy X-ray absorptiometry and dietary habits were assessed with the brief-type self-administered diet history questionnaire. Participants were divided into premenopausal (*n* = 46) and postmenopausal (*n* = 111) groups and the correlation between the BMD Z-score (Z-score) and the intakes of 43 nutrients was investigated separately for each group. In premenopausal women, the daily intake of ash, calcium, and α-tocopherol was positively correlated with the Z-score (Pearson’s correlation coefficient, *R* = 0.31, 0.34, 0.33, *p* = 0.037, 0.020, 0.027, respectively). When dividing the consumption of ash, calcium, and α-tocopherol into low, middle, and high tertiles, the Z-score significantly differed only between the α-tocopherol tertiles. After adjustment for age, body mass index, and lifestyle factors, daily intake of α-tocopherol remained significantly associated with the Z-score (regression coefficient = 0.452, *p* = 0.022). No nutrient was found to be significantly correlated with the Z-score in postmenopausal women. Increase in the intake of α-tocopherol could help maintain bone mass in premenopausal women.

## 1. Introduction

The estimated number of Japanese adults with osteoporosis is 12.8 million, which accounts for about 10% of the entire population [1,2], and the incidence of osteoporosis-related hip fracture is increasing [3]. Osteoporosis and related fractures are becoming more common with age and noticeably accelerating in the postmenopausal period, which is associated with estrogen deficiency. Osteoporosis-induced fragile fractures lead to immobilization, bed rest, and inactivity, impairing various organ functions and leading to physical deconditioning. These outcomes result in long-term activity restriction, increased muscle weakness, and decreased physical strength. Preventing osteoporosis is critical to breaking this vicious cycle and extending healthy life expectancy.

Some minerals and vitamins are considered necessary for bone metabolism. Calcium is the major component of bone and vitamin D is indispensable for calcium absorption from the intestine, and maintenance of calcium homeostasis. Vitamin K also plays a role as a coenzyme in osteocalcin γ-carboxylation, promoting bone calcification. Several epidemiological studies have suggested that calcium and vitamin D supplementation have protective effects on bone loss and fractures [4,5,6,7,8]. The beneficial effects of vitamin K on bone mineral density (BMD) and osteocalcin γ-carboxylation status have been reported in previous studies [9,10,11]. However, the effects of other various nutrients on bone health remain unclear. The aim of the current study was to investigate the association between the dietary consumption of a variety of nutrients and BMD in Japanese middle-aged women.

## 2. Materials and Methods

### 2.1. Study Population

We performed a cross-sectional retrospective study based on the clinical records of Japanese women who enrolled in the Systematic Health and Nutrition Education Program conducted at the menopause clinic of the Tokyo Medical and Dental University, from January 2009 to August 2017. All of the middle-aged women who enrolled in this program had visited our clinic for treatment of menopausal symptoms and provided informed consent for participation in this research. The aim of the program was to improve the physical and psychological health of the women through a comprehensive approach to their general physical and mental health status and lifestyle. Participants received appropriate medications and advice on diet and exercise regimens based on the assessment. During the study period, 229 out of 700 patients underwent dual-energy X-ray absorptiometry (DEXA) and their dietary habits were assessed with a brief-type self-administered diet history questionnaire (BDHQ) with a 6-month interval in-between. Seventy-two women who had been treated with estrogen or anti-osteoporosis drugs were excluded. In the remaining 157 women, the correlations between the BMD Z-score and the intake of nutrients were investigated.

The study protocol was reviewed and approved by the Tokyo Medical and Dental University Review Board. The study was conducted in accordance with the Declaration of Helsinki.

### 2.2. Measurements

Lumbar spine BMD (L2-4) was measured with DEXA (QDR-4500, Hologic Inc. Marlborough, CA, USA). The body composition of the participants, including height, weight, and body mass index (BMI), was measured using a body composition analyzer (MC190-EM, Tanita, Tokyo, Japan). Participants underwent a medical interview for lifestyle factors such as smoking, drinking, and exercise habits. Smoking status was evaluated according to the number of cigarettes smoked per day: none, less than 20, and 20 or more. Drinking status was also assessed according to the frequency of alcohol consumption: none, occasionally, and daily. Participants were additionally evaluated for the regularity of exercise activities: yes and no. Menopausal status of the study participants were also assessed. Postmenopausal status was defined as the absence of menstruation in the past 12 months.

The BDHQ was developed as a short version of the self-administered diet history questionnaire focused on the typical Japanese diet [12]. It evaluates the dietary intake frequency of 58 food items during the past month, which enables a computer algorithm to estimate the daily intake of 96 nutrient factors after adjustment for total calorie intake. BDHQ has previously been validated by comparison with dietary records using a semi-weighed method [13,14]. We investigated the associations between the BMD Z-score in 157 participants and 43 major nutrient factors. Table 1 shows the 43 major nutrients analyzed using the BDHQ. Potential renal acid load (PRAL) and net endogenous acid production (NEAP), the indicators of dietary acid load, were calculated as follows using the data derived from the BDHQ [15,16,17].

### 2.3. Statistical Analyses

Continuous variables are presented as mean ± standard deviation. Pearson’s correlation coefficients (*R*) were calculated between the BMD Z-score and 43 major nutrient factors and dietary acid load. Required sample size was estimated at 85, as calculated from the level of effect size, two-sided alpha, and powers of 0.3, 0.05, and 0.20, respectively. One-way analysis of variance (ANOVA) was used to evaluate the differences in the BMD Z-score between the tertiles, divided by the consumption of selected nutrients. The associations between the BMD Z-score and selected nutrients after adjustment for age, BMI, cigarette smoking, alcohol consumption, and exercise, were examined by multivariate linear regression analysis. |*R*| > 0.3 and *p* < 0.05 were considered statistically significant. Statistical analyses were performed with GraphPad Prism version 5.02 (GraphPad Software, San Diego, CA, USA) and JMP software version 12 (SAS Institute Inc. Cary, NC, USA).

(1)$$\begin{array}{ll} \mathrm{PRAL}\;(\mathrm{mEq}/d) & =\;0.4888\;\times \;\mathrm{protein}\;(g/d)\\ & +\;0.0366\;\times \;\mathrm{phosphorus}\;(\mathrm{mg}/d)\\ & -\;0.0205\;\times \;\mathrm{potassium}\;(\mathrm{mg}/d)\\ & -\;0.0125\;\times \;\mathrm{calcium}\;(\mathrm{mg}/d)\\ & -\;0.0263\;\times \;\mathrm{magnesium}\;(\mathrm{mg}/d) \end{array}$$

(2)NEAP (mEq/d) = [54.5 × protein (g/d) ÷ potassium (mEq/d)] − 10.2

## 3. Results

The mean consumption of 43 nutrients and dietary acid load in the study participants are shown in Table 1. The average age of the participants was 54.5 ± 7.0 years, and BMI was 21.7 ± 3.5 kg/m^2^. Their BMD and BMD Z-scores were 1.044 ± 0.167 g/cm^2^ and 0.15 ± 1.26, respectively (Table 2). We first evaluated the correlation between the BMD Z-score and the daily intake of 43 nutrient factors and dietary acid load in 157 study participants. No significant correlation was observed between the BMD Z-score and the intake of 43 nutrients and dietary acid load (Table 3).

Next, we performed a post hoc subgroup analysis of pre- and postmenopausal women to assess the effects of the intake of nutrients on BMD with and without estrogen exposure. Among 157 women, 46 women were premenopausal and 111 women were postmenopausal. Their mean ages were 49.3 ± 2.9 and 56.6 ± 7.1 years, respectively, and BMIs were 21.5 ± 3.9 and 21.8 ± 3.4 kg/m^2^, respectively. Moreover, BMDs were 1.124 ± 0.181 and 1.010 ± 0.148 g/cm^2^, respectively, and the BMD Z-scores were 0.30 ± 1.35 and 0.09 ± 1.23, respectively in the pre- and post-menopausal groups (Table 2). Among the 43 nutrient factors and dietary acid load that were analyzed, only the daily estimated intake of ash, calcium, and α-tocopherol in premenopausal women had significant positive correlations with the BMD Z-score (ash, *R* = 0.31, *p* = 0.037; calcium, *R* = 0.34, *p* = 0.020; α-tocopherol, *R* = 0.33, *p* = 0.027) (Table 3, Figure 1). No such associations were found between the nutrients and dietary acid load and the BMD Z-score in postmenopausal women.

Furthermore, we divided study participants according to their daily consumption of ash, calcium, and α-tocopherol into three groups: low (*n* = 15), middle (*n* = 15), and high (*n* = 16). There was a significant difference in the BMD Z-score between the α-tocopherol tertiles (*p* = 0.022, one-way ANOVA), but not between the ash and calcium tertiles (mineral, *p* = 0.069; calcium, *p* = 0.129) (Figure 2).

Finally, we performed multivariate linear regression analysis to clarify the relationship between the daily intake of α-tocopherol and the BMD Z-score in premenopausal women (Table 4). Using cut-off values for a variance inflation factor of less than 10, we found no multicollinearity between the independent variables, including age, BMI, smoking, drinking, and exercise. After adjustment for age and BMI (Model 2), and for smoking, drinking, and exercise (Model 3), the daily consumption of α-tocopherol was still significantly correlated with the BMD Z-score (Model 2, regression coefficient (β) = 0.366, *p* = 0.029; Model 3, β = 0.452, *p* = 0.022).

## 4. Discussion

In this cross-sectional analysis, the BMD Z-score was positively correlated with the daily intake of α-tocopherol in premenopausal women, but not in postmenopausal women.

Vitamin E (VE) comprises four tocopherols (TPs) and four tocotrienols; both groups have α-, β-, γ-, δ-isomers. α-TP has the highest bioavailability and the strongest antioxidative activity [18] and is the most used compound in VE supplements. The main dietary sources of VE are plant oils, nuts, seeds, fish, and shellfish. According to the dietary reference intake defined by Ministry of Health, Labor and Welfare in Japan, the adequate intake levels of VE (α-TP) in adult men versus women are 6.5 versus 6.0 mg/day, respectively; the and tolerable upper intake of VE (α-TP) in adult men versus women are 750–900 versus 650–700 mg/day, respectively [19]. In the current study, the highest tertile of α-TP intake was approximately 1.1–1.8 mg/MJ/day, suggesting appropriate daily consumption.

Although the precise pathogenesis of osteoporosis is unclear, oxidative stress is a significant contributor. Both aging and loss of sex hormones are associated with increased levels of reactive oxygen species [20,21,22], leading to (i) decreased osteoblastogenesis; (ii) increased osteoblast and osteocyte apoptosis; and (iii) increased formation and activation of osteoclasts [20,22,23], all of which exert negative effects on bone mass and strength [20,22,24,25]. Oxidative stress also increases advanced glycation end products, which lead to the suppression of bone calcification and decline in bone strength [26,27].

As oxidative stress is one of the central mechanisms in bone loss, VE, a well-known antioxidant, could inhibit the progress of osteoporosis. VE, as a peroxyl radical scavenger, inhibits the propagation of the peroxidative chain and terminates chain reactions [18,28,29]. For example, Roob et al. reported that the oral administration of VE attenuated lipid peroxidation induced by intravenous iron administration in patients receiving hemodialysis [30]. Several epidemiological studies report positive correlations between VE level/intake and BMD. A cross-sectional study involving 232 early postmenopausal women demonstrated that serum VE levels have a positive correlation with the BMD T-score in the lumbar spine [31]. Shi et al. also reported that the dietary intake of VE and serum α-TP concentration were positively associated with lumbar spine, hip, and femur BMD in 2178 women, most of whom were elderly [32]. Additionally, several reports have shown associations between inadequate VE intake and fracture risk. A lower dietary intake of VE in current smokers was associated with an increased risk of hip fracture in a prospective cohort study of 951 postmenopausal women [33]. A recent large-scale longitudinal study of 61,433 elderly women demonstrated that the hip fracture rate exponentially increased when the dietary α-TP intake was less than 5 mg per day [34].

In contrast, several studies have reported negative effects of VE on bone health. A few suggested that α-TP suppresses bone formation in postmenopausal women [35,36], while others showed that the serum level of α-TP was negatively associated with BMD in the elderly population [37]. Additionally, Fujita et al. demonstrated that α-TP promoted osteoclast fusion and multinucleation in vitro and that excessive α-TP intake in rodents induced increased bone absorption and decreased bone mass irrespective of antioxidative effects [38], although the dose used in the animal study greatly exceeded the recommended daily intake of VE for humans. Moreover, it was recently reported that a dietary pattern rich in VE, fats, and fatty acids was negatively associated with BMD in postmenopausal women [39]. On the contrary, Ikegami et al. showed that excessive VE intake was not responsible for bone loss in ovariectomized female mice, regardless of dietary fat content [40]. Further exploration is warranted to determine the effects of VE combined with other nutrients on bone health.

In the present study, α-TP played a beneficial role in BMD in premenopausal women but not in postmenopausal women, implying that the negative effects of aging and estrogen deficiency on bone health might be too potent for the antioxidative effects of α-TP to overcome. We also did not find associations between BMD and nutrients regarded as crucial for bone metabolism, such as calcium, vitamin D, and vitamin K. The median daily intake of these nutrients in this population (calcium, 612.7 mg; vitamin D, 12.3 μg; vitamin K, 368.5 μg) were estimated as adequate according to the dietary reference intake defined by Ministry of Health, Labor and Welfare in Japan (calcium, 550 mg; vitamin D, 5.5 μg; vitamin K, 150 μg), which could be the reason for the lack of associations between BMD and calcium, vitamin D, and vitamin K.

One of the major limitations of our study was that the study population was small, especially in the premenopausal group. As the study population exclusively consisted of Japanese women who visited our menopause clinic, it is difficult to extrapolate our current findings to a wider population. Additionally, we could not determine causal relationships due to the cross-sectional nature of our study. Although we evaluated the daily estimated consumption of nutrients, we did not assess the serum concentration of α-TP. Finally, we did not investigate several influential factors on bone health, such as physical activity level, diabetes, kidney diseases, steroid therapy, and the consumption of other dietary supplements.

Despite these limitations, this study has several strengths and a novel finding. First, the consumption of as many as 43 nutrient factors was simultaneously analyzed using the BDHQ. Relatively young women were included in the current study, enabling us to analyze pre- and postmenopausal women separately. To the best of our knowledge, this is the first study to report the beneficial effect of α-TP on bone health in premenopausal women.

## 5. Conclusions

In conclusion, the dietary intake of α-TP was shown to be positively associated with BMD in premenopausal Japanese women. An increase in the dietary consumption of VE could help maintain the bone mass in premenopausal women.

## Figures and Tables

**Figure 1 nutrients-11-02474-f001:**
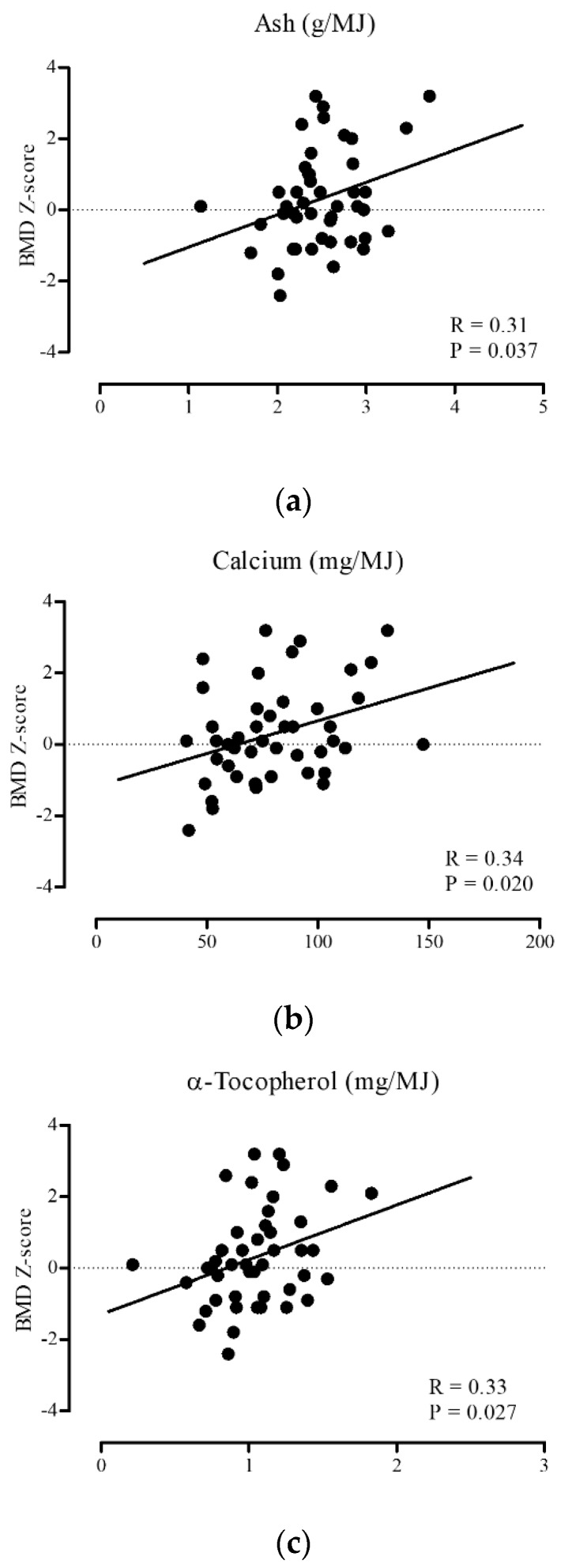
Correlations between BMD Z-score and daily intakes of (**a**) ash, (**b**) calcium, and (**c**) α-tocopherol in premenopausal women. BMD, bone mineral density. Significant positive correlations were found between BMD Z-score and the intake of ash, calcium, and α-tocopherol in premenopausal women.

**Figure 2 nutrients-11-02474-f002:**
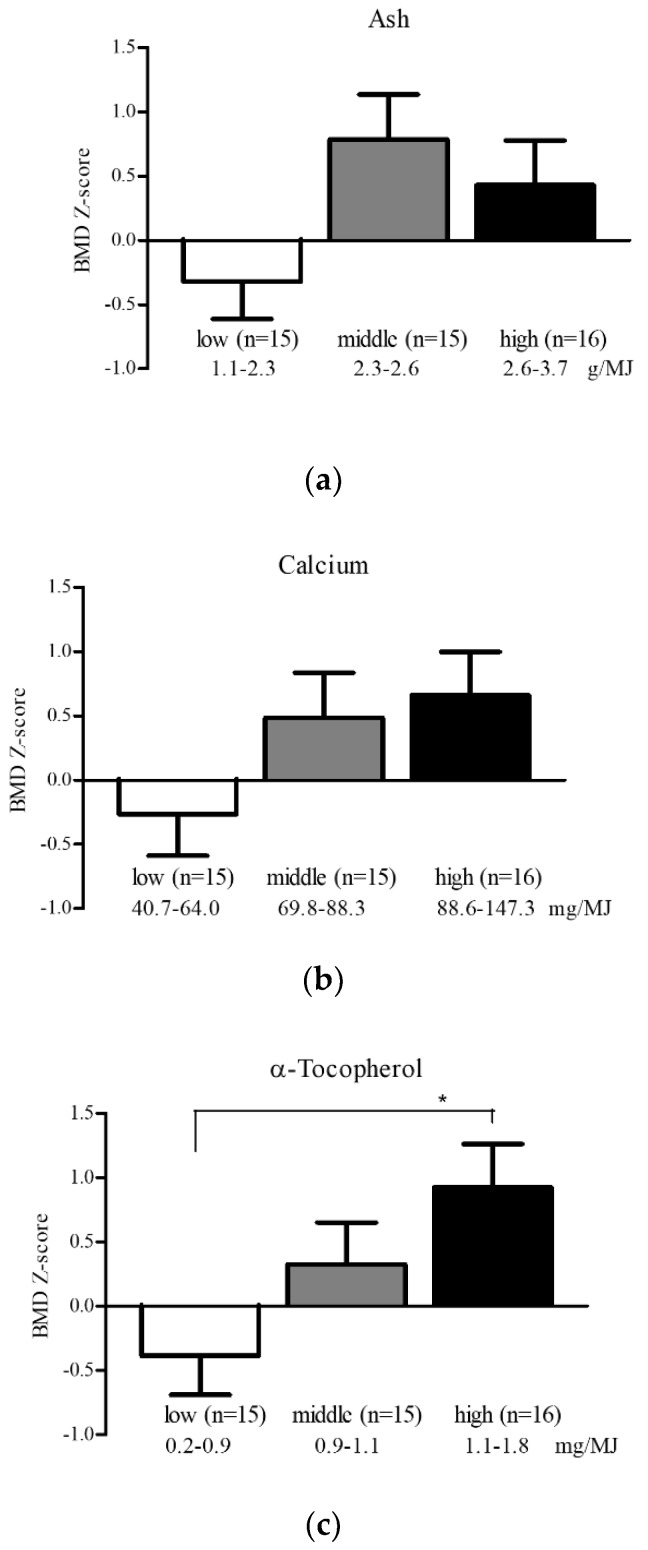
Differences in BMD Z-score by tertiles of (**a**) ash; (**b**) calcium; and (**c**) α-tocopherol intake in premenopausal women. Data are presented as mean and standard error. * *p* < 0.05 versus low consumption group, one-way ANOVA. BMD, bone mineral density; ANOVA, analysis of variance. Significant difference in BMD Z-score was observed between the α-tocopherol tertiles in premenopausal women.

**Table 1 nutrients-11-02474-t001:** Consumption of 43 nutrient factors and dietary acid load in the study participants.

	Study Participants (*n* = 157)	Premenopause (*n* = 46)	Postmenopause (*n* = 111)	*p* Value
Protein, % energy	16.5 (3.4)	15.1 (2.7)	17.0 (3.5)	0.001 **
Animal protein, % energy	9.4 (3.5)	8.2 (3.1)	9.9 (3.5)	0.004 **
Vegetable protein, % energy	7.1 (1.3)	6.9 (1.4)	7.2 (1.3)	0.052
Fat, % energy	26.9 (5.5)	26.9 (6.1)	26.9 (5.3)	0.767
Animal fat, % energy	12.7 (4.2)	12.3 (5.0)	12.9 (3.8)	0.300
Vegetable fat, % energy	14.2 (3.6)	14.6 (3.7)	14.0 (3.5)	0.164
Carbohydrate, % energy	53.5 (8.1)	54.5 (7.7)	53.1 (8.3)	0.313
Ash, g/MJ	2.7 (0.6)	2.5 (0.5)	2.8 (0.6)	0.005 **
Sodium, mg/MJ	575.4 (140.2)	532.7 (115.5)	593.1 (146.1)	0.027 *
Potassium, mg/MJ	411.6 (120.1)	381.9 (104.6)	423.9 (124.3)	0.034 *
Calcium, mg/MJ	88.4 (30.0)	80.1 (25.2)	91.9 (31.2)	0.024 *
Magnesium, mg/MJ	38.9 (9.1)	36.2 (7.6)	40.0 (9.4)	0.017 *
Phosphorus, mg/MJ	153.1 (33.8)	141.4 (26.4)	158.0 (35.4)	0.005 **
Iron, mg/MJ	1.2 (0.3)	1.1 (0.3)	1.3 (0.3)	0.007 **
Zinc, mg/MJ	1.1 (0.2)	1.1 (0.2)	1.2 (0.2)	0.003 **
Copper, mg/MJ	0.2 (0.0)	0.2 (0.0)	0.2 (0.0)	0.006 **
Manganese, mg/MJ	0.5 (0.1)	0.4 (0.1)	0.5 (0.1)	0.010 *
Daidzein, mg/MJ	2.3 (1.3)	2.0 (1.4)	2.5 (1.3)	0.012 *
Genistein, mg/MJ	3.9 (2.3)	3.4 (2.3)	4.2 (2.2)	0.012 *
Retinol, μg/MJ	54.7 (32.6)	57.8 (36.3)	53.4 (31.0)	0.626
β-Carotene equivalents, μg/MJ	689.4 (457.5)	631.4 (496.7)	713.4 (440.3)	0.102
Retinol equivalents, μg/MJ	112.5 (47.1)	110.8 (50.5)	113.2 (45.8)	0.608
Vitamin D, μg/MJ	2.0 (1.3)	1.6 (0.9)	2.1 (1.4)	0.039 *
α-Tocopherol, mg/MJ	1.1 (0.3)	1.0 (0.3)	1.1 (0.3)	0.220
Vitamin K, μg/MJ	53.6 (28.2)	47.9 (25.6)	56.0 (29.0)	0.110
Vitamin B1, mg/MJ	0.1 (0.0)	0.1 (0.0)	0.1 (0.0)	0.019 *
Vitamin B2, mg/MJ	0.2 (0.0)	0.2 (0.0)	0.2 (0.0)	0.004 **
Niacin, mg NE/MJ	2.4 (0.6)	2.3 (0.6)	1.4 (0.7)	0.061
VitaminB6, mg/MJ	0.2 (0.1)	0.2 (0.0)	0.2 (0.1)	0.014 *
VitaminB12, μg/MJ	1.3 (0.6)	1.1 (0.5)	1.4 (0.7)	0.043 *
Folic acid, μg/MJ	56.1 (20.3)	51.2 (18.4)	58.1 (20.7)	0.039 *
Pantothenic acid, mg/MJ	1.0 (0.2)	0.9 (0.2)	1.0 (0.2)	0.008 **
Vitamin C, mg/MJ	20.3 (9.2)	18.2 (8.4)	21.2 (9.5)	0.064
Saturated fatty acid, g/MJ	1.8(0.5)	1.8 (0.6)	1.8 (0.4)	0.531
Monounsaturated fatty acid, g/MJ	2.2 (0.5)	2.2 (0.5)	2.2 (0.5)	0.756
Polyunsaturated fatty acid, g/MJ	1.6 (0.3)	1.5 (0.3)	1.6 (0.3)	0.737
Cholesterol, mg/MJ	49.0 (15.7)	45.7 (14.5)	50.3 (16.0)	0.118
*n*-3 fatty acid, g/MJ	0.3 (0.1)	0.3 (0.1)	0.3 (0.1)	0.031 *
*n*-6 fatty acid, g/MJ	1.2 (0.3)	1.2 (0.3)	1.2 (0.3)	0.873
Soluble dietary fiber, g/MJ	0.5 (0.2)	0.5 (0.2)	0.5 (0.2)	0.073
Insoluble dietary fiber, g/MJ	1.4 (0.5)	1.3 (0.4)	1.4 (0.5)	0.089
Dietary fiber, g/MJ	2.0 (0.7)	1.8 (0.6)	2.0 (0.7)	0.078
Alcohol, g/MJ	0.6 (1.5)	0.7 (1.2)	0.6 (1.6)	0.053
Potential renal acid load, mEq/d	−0.3 (14.5)	−0.2 (12.9)	−0.3 (15.1)	0.989
Net endogenous acid production, mEq/d	43.3 (12.0)	42.5 (11.2)	43.5 (12.4)	0.626

* *p* < 0.05, ** *p* <0.01, values are mean (standard deviation).

**Table 2 nutrients-11-02474-t002:** Background characteristics of the participants.

	Study Participants	Premenopause	Postmenopause
	(*n* = 157)	(*n* = 46)	(*n* = 111)
Age, years	54.5 (7.0)	49.2 (2.9)	56.6 (7.1)
Height, cm	156.8 (6.1)	158.1 (5.2)	156.2 (6.4)
Weight, kg	53.2 (8.2)	53.5 (8.9)	53.0 (7.9)
Body mass index, kg/m^2^	21.7 (3.5)	21.5 (3.9)	21.8 (3.4)
Bone mineral density, g/cm^2^	1.044 (0.167)	1.124 (0.181)	1.010 (0.148)
Bone mineral density Z-score	0.15 (1.26)	0.30 (1.35)	0.09 (1.23)
Drinking, %			
never/sometimes/daily	59.9/31.2/8.9	45.7/45.7/8.6	65.8/25.2/9.0
Smoking, %			
none/less than 20/20 or more cigarettes per day	89.8/6.4/3.8	84.8/8.7/6.5	91.9/5.4/2.7
Exercise, %			
yes/no	55.3/44.7	50.0/50.0	57.5/42.5

**Table 3 nutrients-11-02474-t003:** Correlations between 43 nutrient factors and dietary acid load and the bone mineral density (BMD) Z-score in the study participants.

	Study Participants		Premenopause		Postmenopause	
	*R*	*p*	*R*	*p*	*R*	*p*
Protein	0.12	0.133	0.10	0.498	0.10	0.201
Animal protein	0.11	0.156	0.07	0.622	0.09	0.277
Vegetable protein	<0.01	0.913	0.03	0.843	0.04	0.624
Fat	0.1	0.214	0.15	0.323	0.04	0.599
Animal fat	0.13	0.097	0.11	0.469	0.12	0.141
Vegetable fat	<−0.01	0.983	0.09	0.527	−0.06	0.457
Carbohydrate	−0.08	0.306	−0.08	0.594	−0.05	0.569
Ash	0.21	0.007	0.30	0.042	0.15	0.057
Sodium	0.2	0.012	0.20	0.175	0.14	0.070
Potassium	0.16	0.039	0.27	0.064	0.13	0.096
Calcium	0.15	0.059	0.33	0.023	0.05	0.545
Magnesium	0.15	0.053	0.23	0.123	0.10	0.189
Phosphorus	0.14	0.072	0.22	0.132	0.08	0.318
Iron	0.14	0.082	0.19	0.208	0.14	0.086
Zinc	0.05	0.526	0.05	0.745	0.0	0.412
Copper	0.08	0.315	0.08	0.593	0.09	0.250
Manganese	0.12	0.151	0.07	0.627	0.16	0.042
Daidzein	0.04	0.59	0.07	0.620	0.01	0.921
Genistein	0.04	0.588	0.08	0.604	0.01	0.918
Retinol	−0.06	0.485	−0.07	0.630	−0.07	0.366
β-Carotene equivalents	0.11	0.165	0.27	0.067	0.09	0.284
Retinol equivalents	0.05	0.523	0.16	0.285	0.01	0.905
Vitamin D	0.11	0.14	0.10	0.499	0.06	0.447
α-Tocopherol	0.17	0.031	0.33	0.025	0.10	0.227
Vitamin K	0.1	0.228	0.13	0.371	0.06	0.443
Vitamin B1	0.13	0.095	0.23	0.122	0.15	0.064
Vitamin B2	0.16	0.041	0.16	0.281	0.13	0.098
Niacin	0.11	0.16	−0.01	0.927	0.13	0.107
Vitamin B6	0.16	0.045	0.11	0.477	0.19	0.014
Vitamin B12	0.11	0.176	<0.01	0.985	0.08	0.327
Folic acid	0.14	0.075	0.20	0.169	0.14	0.081
Pantothenic acid	0.12	0.13	0.11	0.446	0.11	0.179
Vitamin C	0.19	0.016	0.27	0.064	0.19	0.016
Saturated fatty acid	0.12	0.142	0.16	0.294	0.07	0.376
Monounsaturated fatty acid	0.06	0.442	0.09	0.558	0.03	0.732
Polyunsaturated fatty acid	0.05	0.527	0.14	0.356	−0.03	0.665
Cholesterol	0.09	0.249	0.15	0.320	0.04	0.587
*n*-3 fatty acid	0.11	0.177	0.05	0.753	0.06	0.426
*n*-6 fatty acid	0.02	0.786	0.15	0.309	−0.07	0.405
Soluble dietary fiber	0.1	0.219	0.16	0.282	0.11	0.185
Insoluble dietary fiber	0.14	0.074	0.21	0.148	0.15	0.059
Dietary fiber	0.13	0.103	0.20	0.172	0.14	0.073
Alcohol	−0.01	0.869	−0.03	0.817	−0.01	0.870
Potential renal acid load	−0.11	0.178	−0.18	0.229	−0.08	0.394
Net endogenous acid production	−0.13	0.108	−0.21	0.167	−0.09	0.326

*R*, Pearson’s correlation coefficients; BMD, bone mineral density. There was no significant correlation between the BMD Z-score and the intake of 43 nutrients and dietary acid load in 157 study participants. The daily intake of ash, calcium, and α-tocopherol in premenopausal women was positively correlated with the BMD Z-score. No significant correlation was observed between the intake of nutrients and the BMD Z-score in postmenopausal women.

**Table 4 nutrients-11-02474-t004:** Multivariate analysis: the association between daily intake of α-tocopherol and the BMD Z-score.

	β	SE	*p*-Value	*R*2
Model 1	0.366	0.158	0.027	0.107
Model 2	0.366	0.160	0.029	0.178
Model 3	0.452	0.184	0.022	0.270

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
