# Peer review of "Bone Mineral Density in Premenopausal Women Is Associated with the Dietary Intake of α-Tocopherol: A Cross-Sectional Study"

_nutrients, 2019, doi:10.3390/nu11102474_

Round 1

Reviewer 1 Report

Would be better to include a table showing nutrient intakes of the study participants for pre- and post-menopause.

As indicated in the discussion, oxidative stress is important for bone loss, please include the correlation of intake of total carotenoids, an indicator of fruits and vegetables.

Would be better to include the correlation of dietary acid load with BMD too.

Reviewer 2 Report

Interesting findings, please see my suggestions Method: Why did you use Z-scores for the postmenopausal women cohort, I would have presumed the use of T-scores to be a more appropriate outcome variable. Discussion: I would have thought you would cite a more recent publication by Ilesanmi-Oyelere et al. (2019) The relationship between nutrient patterns and bone mineral density in postmenopausal women. I suggest you discuss the results of this paper in comparison to your findings.
